# Field Testing of Porous Pavement Performance on Runoff and Temperature Control in Taipei City

**Yung-Yun Cheng [1], Shang-Lien Lo [1], Chia-Chun Ho [2], Jen-Yang Lin [3],\* and Shaw L. Yu [4]**

1   Graduate Institute of Environmental Engineering, National Taiwan University, Taipei 10617, Taiwan; joanne@fenri.com.tw (Y.-Y.C.); sllo@ntu.edu.tw (S.-L.L.)
2   Department of Civil and Construction Engineering, National Taiwan University of Science and Technology, Taipei 10607, Taiwan; cchocv@mail.ntust.edu.tw
3   Department of Civil Engineering, National Taipei University of Technology, Taipei 10608, Taiwan
4   Department of Civil and Environmental Engineering, University of Virginia, Charlottesville, VA 22903, USA; sly@virginia.edu
\*   Correspondence: jylin@ntut.edu.tw; Tel.: +886-2-2771-2171 (ext. 2664)

**Abstract:** The Taipei University of Technology, under contract from the Taipei City Government, completed a study on porous asphalt (PA) and permeable interlocking concrete brick (PICB) pavement performance with respect to stormwater runoff reduction and surface temperature mitigation. Additionally, the variation of infiltration rates against time of these pavements was monitored. The results show the following: (a) Runoff peak reduction ranged from 16% for large, intense storms to 55% for small, long-duration storms. Rainfall volume reduction ranged from 16% to 77% with an average of 37.6%; (b) Infiltration rate: for PICB, it decreased by 25% to 50% over a 15-month monitoring period, but the rate at one location increased significantly after cleaning; for PA, the rate remained high at one location, but decreased by 70%–80% after 10 months at two other locations, due mainly to clogging problems; (c) Surface temperature: during storm events, porous concrete bricks had on average lower temperatures compared to regular concrete with a maximum difference of 6.6 °C; for porous asphalt the maximum drop was 3.9 °C. During dry days, both PA and PICB showed a tendency of faster temperature increase as the air temperature rose, but also faster temperature decreases as the air cooled when compared to regular pavements. On very hot days, much lower surface temperatures were observed for porous pavements (for PA: 17.0 °C and for PICB: 14.3 °C) than those for regular pavements. The results suggest that large-scale applications of porous pavements could help mitigate urban heat island impacts.

**Keywords:** porous asphalt pavement; permeable concrete bricks; performance assessment; runoff reduction; temperature mitigation; infiltration rates

## 1. Introduction

Porous pavement is considered a cost-effective stormwater management practice that absorbs, treats, and/or stores runoff in highly urbanized areas [1]. The commonly used types of permeable pavement include porous concrete, porous asphalt, and permeable interlocking pavers, although innovative efforts in recent years have been made to try to find new materials that provide better performance and lower costs.

Performance and positive/negative aspects of porous pavements as a stormwater or low impact development (LID) best management practice (BMP) have been documented [2,3]. However, comprehensive and longer term field tests, especially with respect to surface temperature effects, have been relatively few [4]. One notable study was the USEPA Edison Environment Center's tests of three

types of permeable pavements: permeable interlocking concrete paver (PICP); pervious concrete (PC), and porous asphalt (PA). The results show that clogging might affect the infiltration rates at specific locations where solids tend to accumulate. Additionally, about 5%–7% of the runoff infiltrated into the material later evaporates, which could help mitigate the urban heat-island effect [1], as stipulated by an earlier field test study in Arizona [5]. Another comprehensive study by the US Geological Survey [6] observed higher pollutant removal by PC but also a declining infiltration rate. Permeable pavements could reduce the "black ice" effect because water stored in the pores have temperatures (much water at different times, so many different temperatures) above freezing during winter weather.

The transportation sector may potentially be a major user of permeable pavements. For example, the Virginia Department of Transportation (VDOT) recently completed a study on PA application at a Park and Ride facility [7]. Results show that the infiltration rates remained high during a four-year monitoring period, and that the maintenance cost was low (less than $1500 per year). Another recent study by [8] reported significant benefits of PICP in reducing runoff peaks and urban flooding duration through field tests conducted in Canada. Recognizing the important role of permeable pavements in helping protect the environment under the global climate change scenario, the Environment and Water Resources Institute (EWRI) of the American Society of Civil Engineers (ASCE) charged its Permeable Pavement Task Committee to prepare and publish a current state-of-the-art report on permeable pavement technology, including future research needs [9].

The Taipei City Government, as part of its effort in making Taipei into a "Resilient and Adaptive City" under climate change scenarios, has in recent years been implementing the "sponge city" approach in managing stormwater runoff. The approach, similar to that of sustainable urban drainage (SUD), calls for the use of "natural, on-site" treatment of stormwater runoff through the processes of infiltration, detention, storage, etc. [10,11] The Taipei University of Technology was contracted in 2017 to conduct a demonstration study of the performance of porous pavement in order to provide design guidelines and maintenance information needed for the planned large scale implementation of bicycle lanes and pedestrian walkways throughout the city [12].

The main goal of the present study is to collect and analyze field data for a permeable pavement site at the Da-An Vocational High School (DAVHS) in Taipei City. Two types of pavements were installed, namely, a porous asphalt (PA) bicycle lane and a permeable interlocking concrete brick (PICB) pedestrian walkway. The objectives are to assess (a) the stormwater runoff control performance such as peak runoff and volume reduction; (b) the time variation of infiltration rates of pavements, and importantly (c) the surface temperature variations of pavements during storm events and also after relatively long dry periods.

## 2. Materials and Methods

### 2.1. Site Description

A stretch of pedestrian walkway and a bicycle lane, each 200 m long, were constructed in front of the DAVHS by using porous concrete bricks and porous asphalt pavement, respectively. Figure 1 depicts the site location in Taipei City. The bicycle lane is parallel to a high-traffic road constructed with regular asphalt and concrete, which was used as a control. Underneath the walkway, two runoff storage tanks, each with dimensions of 70 m long, 2 m wide, and 0.5 m deep (70 $m^3$ in volume) were constructed to store infiltration runoff water. Figure 2 is a picture of the test site.

Figure 3 shows the installation of the underground stormwater storage tanks. A cross-section of the storage tank design with materials used is shown in Figure 4.

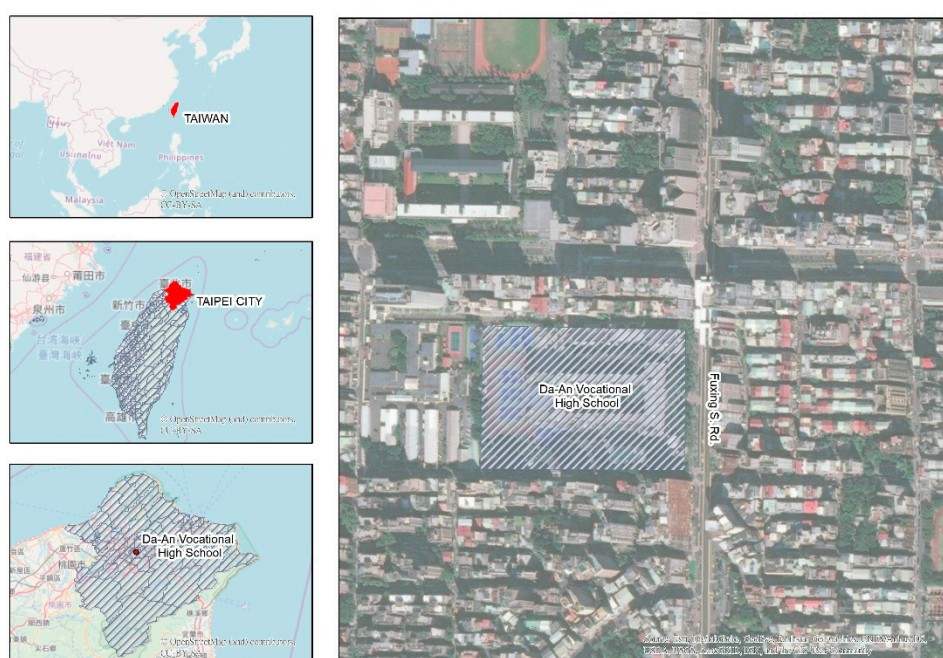

**Figure 1.** The test site location in Taipei City.

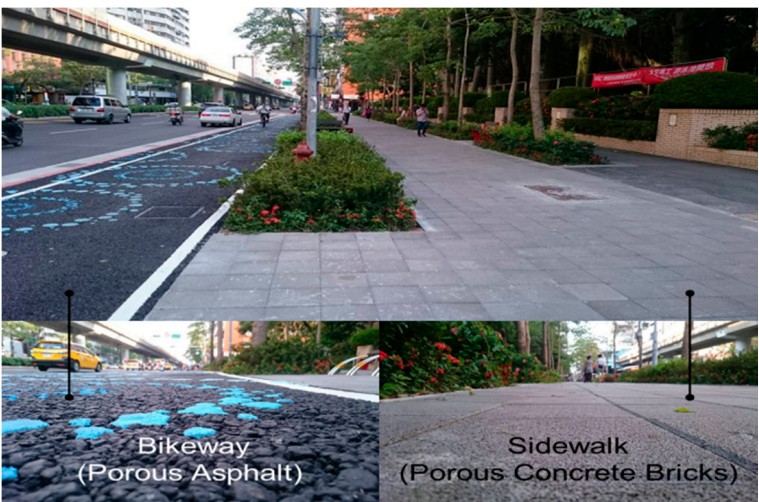

**Figure 2.** A picture of the permeable pavements test site.

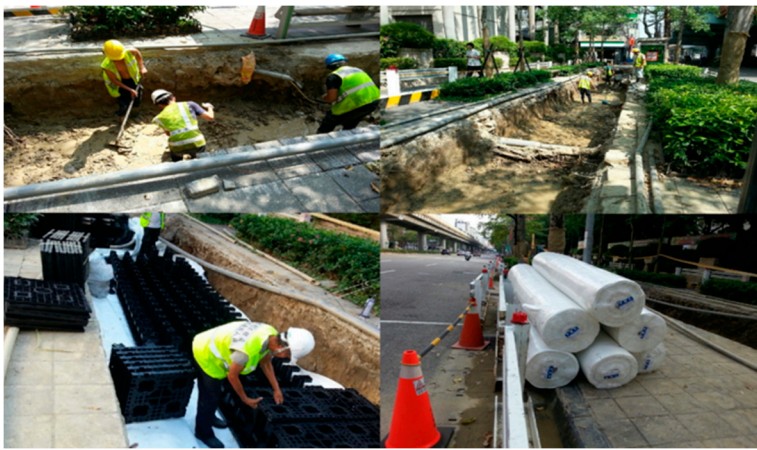

**Figure 3.** Installation of the underground stormwater storage tanks.

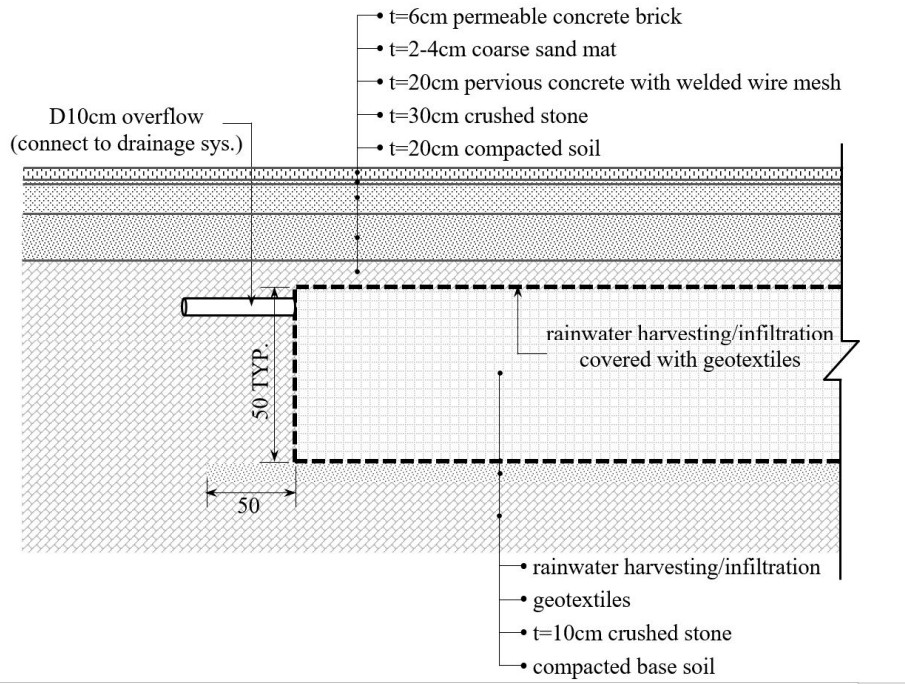

**Figure 4.** Cross-section of the underground rainwater storage system.

## 2.2. Monitoring Plan

The following describes the type of data collected at the site and the equipment used:

- Rainfall: One rain gage (HOBO-ON SET self-recording gage) on the roof of a campus security room at the site. Rainfall measured at 10 min intervals.
- Runoff stored at the two underground runoff storage tanks: Two automatic water level recorders. Water level measured at hourly intervals.
- Flow measurements at four locations: Rectangular weirs at 10 min intervals.

Infiltration rates at seven locations were measured on a monthly basis using permeameters, one of which is shown in Figure 5. The measurement process involved first filling the pipe with water, recording the height of the water level drop in the pipe per unit time, and calculating the infiltration rate at various times. The variation in infiltration rate was plotted over time, and the permeability coefficient (k) of permeable pavement was defined when the infiltration rate reached stability.

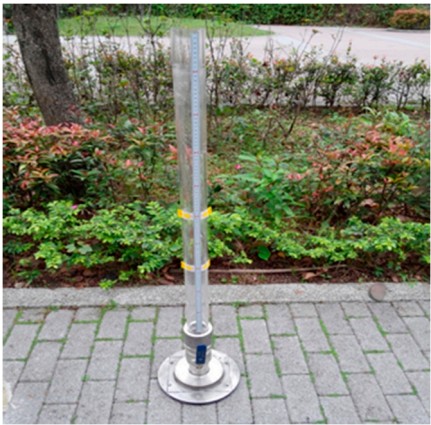

**Figure 5.** A permeameter.

Surface temperatures were measured at six locations during storm events and on days after relatively long dry periods from 9 am to 9 pm at 10 min intervals. HOBO MX2300 Bluetooth recorders were used.

Ground water levels at an observation well were measured by using an automatic water level recorder at hourly intervals.

A detailed description of the monitoring locations for the parameters listed above is shown in Figure 6.

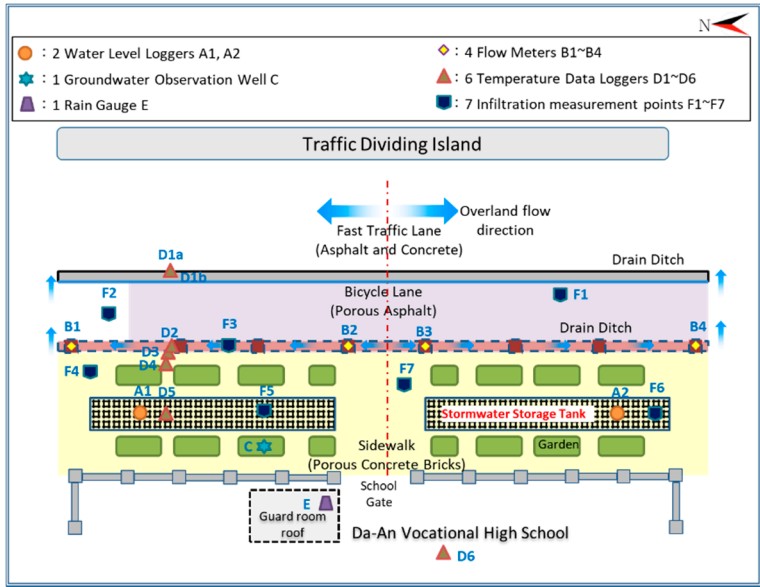

**Figure 6.** Field test site monitoring locations.

It should be noted that, as shown in Figure 6, the surface drainage dividing line (red dotted line) was at the center of the north–south regular traffic lanes and the bicycle lanes. The sampling points were placed so that data were collected on both sides, as described below:

(a) One rain gage on top of the guard room roof—E
(b) Two water level recorders, one each at the underground stormwater tanks—A1, A2
(c) Four flow meters—north side B1 and B2, south side B3 and B4
(d) One groundwater level recorder—C
(e) Six surface temperature recorders—D1 at fast traffic lane with regular pavement; D1a for regular asphalt and D1b for regular concrete section; D2 and D3 for porous asphalt bicycle lanes; D4 and D5 for permeable interlocking concrete brick walkways; and D6 for regular concrete (stucco washing finish) inside the school.
(f) Seven locations for infiltration rate measurements—F1 (south), F2 (north), and F3 (north on top of the drainage ditch) for porous asphalt bike lanes; F4 (north), F5 (north on top of underground stormwater tank); F6 (south on top of underground stormwater tank), and F7 (south) for PICB walkways.

Monitoring of infiltration rates started in February 2018 and ended in May 2019. All other parameters were monitored during the period of April 2018 and May 2019.

To estimate the runoff reduction performance of the porous pavement, a widely used storm water management model (SWMM) was used. The model was calibrated and verified using local data obtained through this study (Figure 7). Details of the SWMM and how it was applied are described in Lin et al. [10].

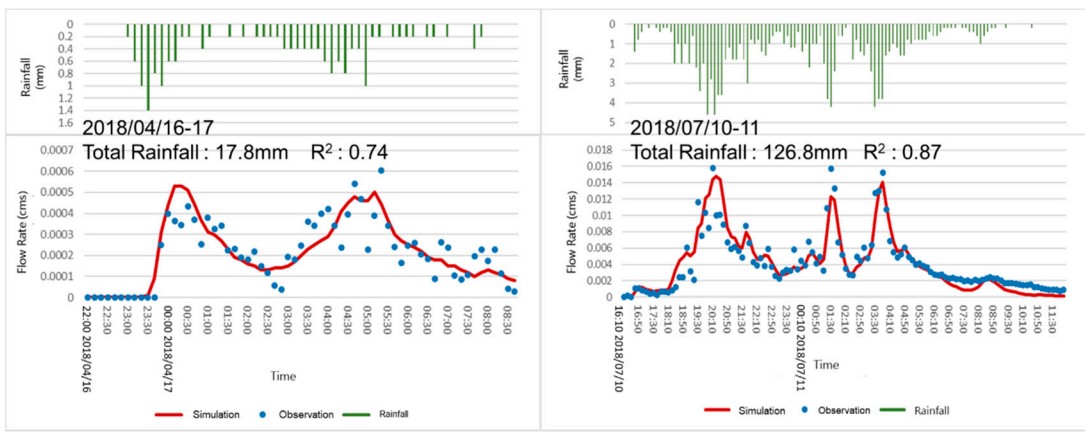

**Figure 7.** Calibration and validation of SWMM model.

## 3. Results and Discussion

### 3.1. Rainfall Events

A total of 36 effective rainfall events were monitored. The screening criteria for an effective storm event followed those described by [13]. Table 1 lists the monitored storm events with their pertinent characteristics such as total rainfall volume, duration, etc.

**Table 1.** Storm events sampled (April 2018–May 2019).

| Date | Total Rainfall (mm) | Rainfall Duration (H:min) | Date | Total Rainfall (mm) | Rainfall Duration (H:min) |
|---|---|---|---|---|---|
| 12 April 2018 | 35.5 | 01:10 | 10,11 October 2018 | 16.8 | 13:05 |
| 16,17 April 2018 | 17.8 | 08:40 | 11,12 October 2018 | 25.8 | 17:45 |
| 8 May 2018 | 26.6 | 04:00 | 16,17 October 2018 | 30.8 | 11:00 |
| 30 May 2018 | 18.2 | 00:30 | 1,2 November 2018 | 21.8 | 42:10 |
| 10,11 July 2018 | 126.8 | 16:10 | 23,24 December 2018 | 41.2 | 30:40 |
| 21 July 2018 | 26.6 | 07:20 | 16,17 January 2019 | 21.1 | 25:00 |
| 24 July 2018 | 27.8 | 02:00 | 23,24 February 2019 | 57.4 | 22:40 |
| 11 August 2018 | 30.2 | 07:50 | 6–10 March 2019 | 89.4 | 84:20 |
| 12 August 2018 | 45.8 | 06:50 | 25 March 2019 | 15.0 | 06:10 |
| 17 August 2018 | 31.2 | 01:40 | 29,30 March 2019 | 24.0 | 05:20 |
| 20 August 2018 | 13.0 | 00:40 | 11 April 2019 | 15.6 | 08:30 |
| 30 August 2018 | 54.4 | 09:20 | 15,16 April 2019 | 29.0 | 24:00 |
| 1 September 2018 | 22.6 | 01:10 | 20,21 April 2019 | 40.6 | 10:40 |
| 7 September 2018 | 39.6 | 03:50 | 22 April 2019 | 23.8 | 05:00 |
| 8 September 2018 | 152.0 | 06:00 | 1 May 2019 | 46.6 | 17:10 |
| 9 September 2018 | 18.4 | 17:20 | 2,3 May 2019 | 14.4 | 20:30 |
| 15,16 September 2018 | 20.4 | 27:30 | 6,7 May 2019 | 24.2 | 27:10 |
| 25–27 September 2018 | 24.2 | 35:00 | 9,10 May 2019 | 14.0 | 08:20 |

During the monitoring period, one "extra-large" storm (defined as >200 mm/d, or >100 mm/3 h) of 152 mm on 8 September 2018, and one "large" storm (defined as >80 mm/d, or >40 mm/h) of 126.8 mm on 10 July 2018, were observed. Most other events were from small to median sizes. Rainfall duration varied from 30 min to over 84 h. The maximum rainfall intensity was observed on 12 April 2018 (35 mm/h), and the lowest intensity storm occurred on 6 May 2019 with an intensity of 2.6 mm/h.

### 3.2. Groundwater and Stormwater Storage Tank Levels

Groundwater levels, measured at hourly intervals, during the monitoring period did not show much variation at all, and were always below the bottoms of the underground storage tanks. This affirms that the storage tanks should have the entire volumes available for storing stormwater runoff. On the other hand, the two storage tanks showed increases in water levels and were even at the full-tank level when the rainfall amount reached the "large" storm category, as depicted in Figure 8.

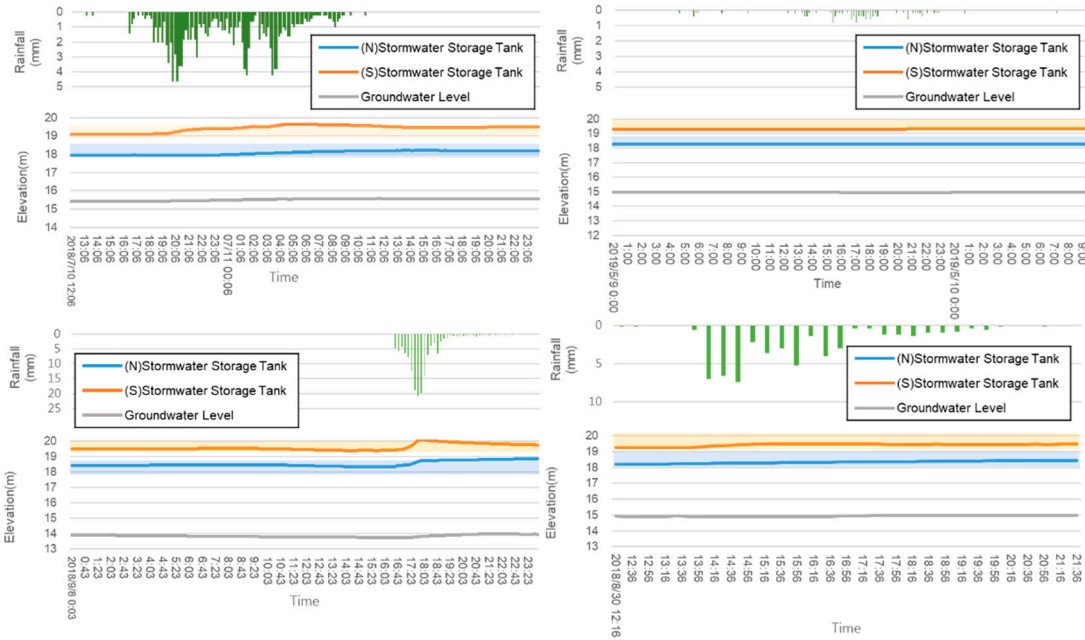

**Figure 8.** Groundwater and rain water storage tank level observations.

Results of the storm-to-storm underground storage tank performance are listed in Table 2. During the 15-month monitoring period, the tanks were 100% full only five times collectively, with the south-side tank three times and the north-side twice. The average stored-capacity rating was 28% for the north tank and 44% for the south. Overall, the tanks did not perform as expected. The reasons might be twofold: one is that the tanks were over-designed size-wise. Given the drainage pattern at the site, overland runoff would move very fast into the drain ditches and then into the city sewers, leaving smaller amounts to slowly infiltrate into the tanks. The other reason is that for higher capture rates, surface runoff should be kept within an "infiltrating" area that feeds the underground tank, and not allowed to runoff quickly.

**Table 2.** Rainwater storage tank performance.

| Date | Total Rainfall (mm) | (North) Fullness (%) | | (South) Fullness (%) | |
|---|---|---|---|---|---|
| | | **Before** | **After** | **Before** | **After** |
| 12 April 2018 | 35.5 | 0 | 1 | 0 | 16 |
| 16,17 April 2018 | 17.8 | 0 | 0 | 0 | 8 |
| 8 May 2018 | 26.6 | 0 | 13 | 0 | 0 |
| 30 May 2018 | 18.2 | 0 | 0 | 0 | 0 |
| 10,11 July 2018 | 126.8 | 0 | 21 | 0 | 100 |
| 21 July 2018 | 26.6 | 0 | 0 | 1 | 20 |
| 24 July 2018 | 27.8 | 0 | 0 | 11 | 36 |
| 11 August 2018 | 30.2 | 0 | 14 | 5 | 81 |
| 12 August 2018 | 45.8 | 17 | 36 | 83 | 87 |
| 17 August 2018 | 31.2 | 0 | 10 | 43 | 85 |
| 20 August 2018 | 13.0 | 13 | 38 | 39 | 71 |
| 30 August 2018 | 54.4 | 18 | 68 | 13 | 81 |
| 1 August 2018 | 22.6 | 18 | 47 | 0 | 44 |
| 7 September 2018 | 39.6 | 21 | 61 | 5 | 65 |
| 8 September 2018 | 152.0 | 44 | 100 | 43 | 100 |
| 9 September 2018 | 18.4 | 100 | 100 | 100 | 100 |
| 15,16 September 2018 | 20.4 | 47 | 59 | 15 | 41 |
| 25–27 September 2018 | 24.2 | 59 | 71 | 12 | 1 |

**Table 2.** *Cont.*

| Date | Total Rainfall (mm) | (North) Fullness (%) | | (South) Fullness (%) | |
|---|---|---|---|---|---|
| | | **Before** | **After** | **Before** | **After** |
| 10,11 October 2018 | 16.8 | 0 | 0 | 24 | 30 |
| 11,12 October 2018 | 25.8 | 0 | 0 | 28 | 55 |
| 16,17 October 2018 | 30.8 | 10 | 0 | 16 | 50 |
| 1,2 November 2018 | 21.8 | 0 | 1 | 12 | 21 |
| 23,24 December 2018 | 41.2 | 0 | 41 | 0 | 35 |
| 16,17 January 2019 | 21.1 | 0 | 5 | 14 | 4 |
| 23,24 February 2019 | 57.4 | 3 | 8 | 5 | 61 |
| 6–10 March 2019 | 89.4 | 0 | 33 | 0 | 42 |
| 25 March 2019 | 15.0 | 0 | 0 | 9 | 38 |
| 29,30 March 2019 | 24.0 | 0 | 0 | 35 | 59 |
| 11 April 2019 | 15.6 | 0 | 34 | 20 | 38 |
| 15,16 April 2019 | 29.0 | 48 | 72 | 33 | 62 |
| 20,21 April 2019 | 40.6 | 0 | 0 | 40 | 92 |
| 22 April 2019 | 23.8 | 0 | 23 | 50 | 95 |
| 1 May 2019 | 46.6 | 17 | 28 | 0 | 42 |
| 2,3 May 2019 | 14.4 | 25 | 47 | 32 | 37 |
| 6,7 May 2019 | 24.2 | 36 | 46 | 13 | 32 |
| 9,10 May 2019 | 14.0 | 35 | 36 | 24 | 36 |

*3.3. Surface Temperature Comparison*

As stated earlier in this paper, since published data on detailed, short-time interval surface temperature variation for porous pavement during wet and dry days are relatively scarce, the present study aimed at collecting such information as a very importance part of the field study effort. Temperature data were collected at six locations, as described before, during all storm events and also during selected dry days after long periods with no rain. Results from these two types of sampling are presented below.

3.3.1. Temperatures during Storm Events

For porous asphalt, data taken at the bicycle lane showed that the surface temperatures for PA were consistently lower than those for regular asphalt pavement. A similar trend was observed for the permeable interlocking concrete bricks. Compared with regular materials, PA had a maximum surface temperature drop of 3.9 °C in summer and 0.7 °C in winter. For PICB the corresponding numbers were 6.6 °C and 0.6 degrees, respectively. The results are shown in Figure 9.

3.3.2. Temperatures during Dry Periods

The data were taken after a prolonged dry period. A typical observation is shown in Figure 10. The 7 August 2018 data were taken after 13 dry days, and the 18 February 2019 data were taken after 1 month of no rain. All observations were made between 9 am and 9 pm at 10 min intervals on the sampling day.

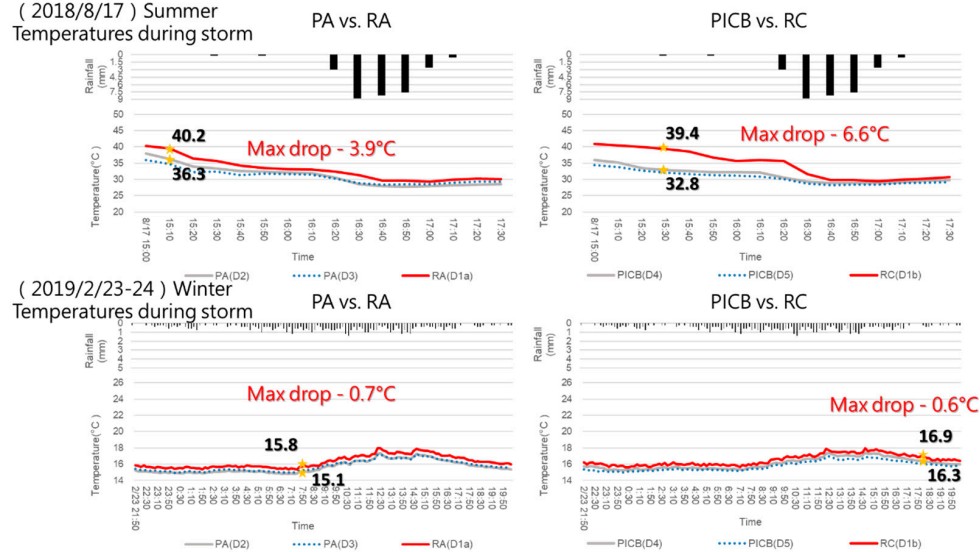

**Figure 9.** Surface temperature during storm events.

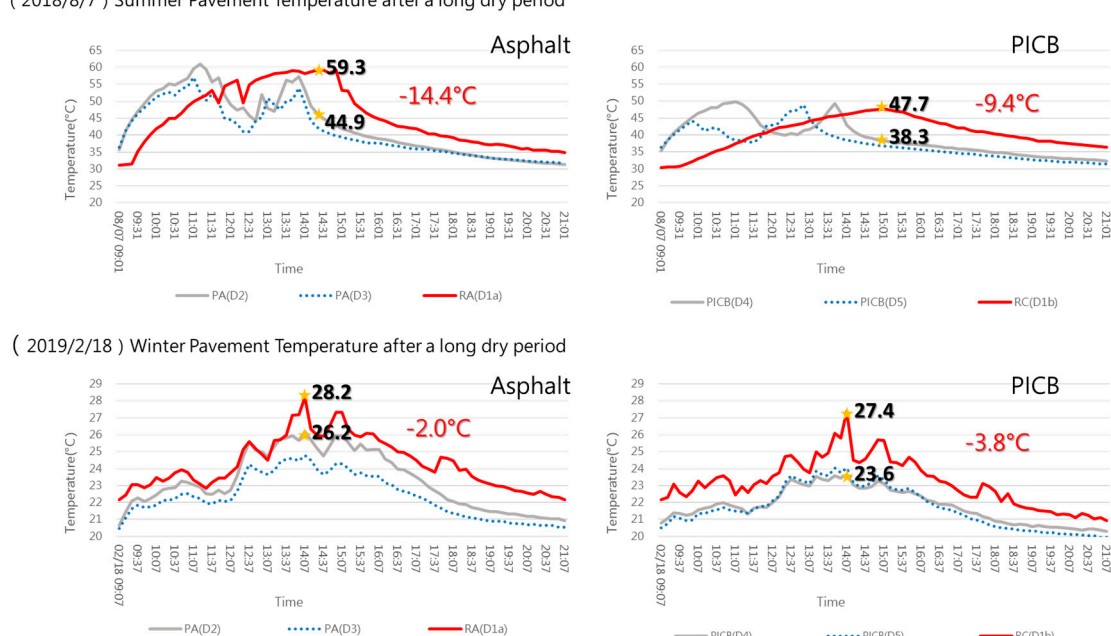

**Figure 10.** Typical surface temperature results during dry periods.

The results indicate that during the winter time, both PA and PICB tended to have lower surface temperatures compared to those for regular materials. In the summer time, however, both porous pavements tended to show faster temperature rise as the air warmed up, but they also showed faster temperature drop as air began to cool down. The temperature drop observed on 7 August 2018 was 14.4 °C for PA and 9.4 °C for PICB, as shown in Figure 10.

The maximum temperature differences between regular and porous pavements observed during prolonged dry and high air temperature periods were observed on 14 September 2018. As shown in Figure 11, the temperature differences were observed as 17.0 °C lower for PA and 14.3 °C lower for PICB.

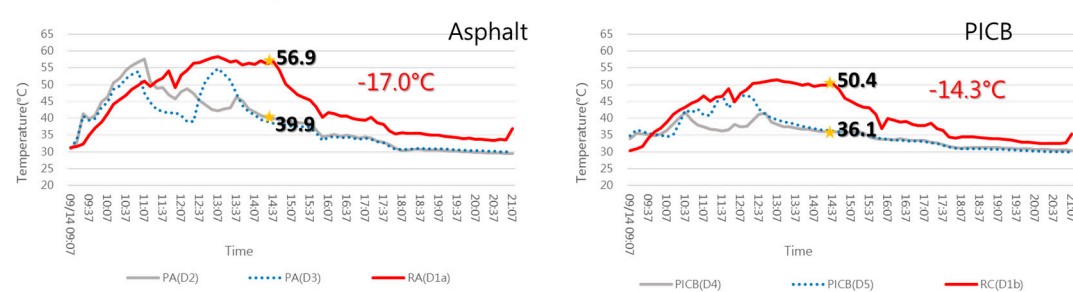

**Figure 11.** Surface temperature during dry periods—maximum values observed.

A composite diagram depicting the continuous time variation of surface temperature for regular asphalt (RA), porous asphalt (PC), and permeable interlocking concrete brick (PICB) pavements is shown in Figure 12. The time series data show clearly that PA and PICB had lower temperatures compared to RA, especially during the high-temperature seasons of summer and fall. The temperature drop was not obvious for PA after January 2019 because the bicycle lane was modified to make the surface smoother in response to bicyclists complains.

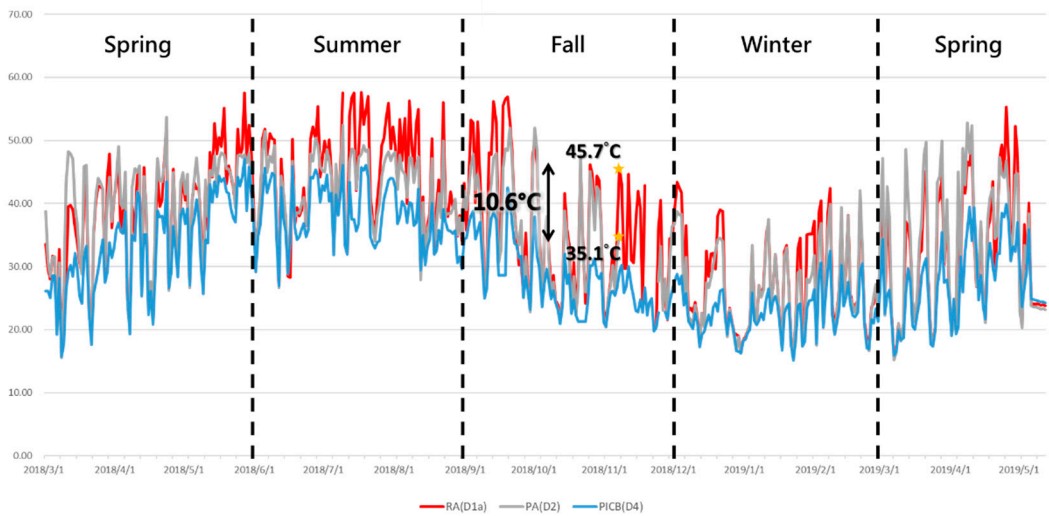

**Figure 12.** Time series of noon-temperature data.

During dry summer days when the air temperatures are in the high thirties °C, the regular asphalt road surface temperatures can reach 65 °C and above. Porous asphalt pavements have been reported to have higher daytime surface temperature but lower night time readings [5]. Data obtained in the present study showed a similar trend. For urban areas in regions with summer–fall rainy seasons, the overall benefits of lower temperatures for porous pavements should be considered. With less heat storage capacity, porous pavement should help mitigate the urban heat island impacts.

In summary, porous pavements help reduce the ground surface temperature, especially during high-temperature periods and in wet weather. During dry weather, porous pavements might show a faster temperature rise when air temperature warms up, but they also show a faster temperature drop as air temperature begins to drop. The overall surface temperature for porous pavements in this case was still lower than those for regular pavements. One notable fact is that although dark asphalt has poor solar-reflective properties, making asphalt porous could lead to the cooling of surfaces by 10 °C or more, which would help mitigate the heat-island effect in urban areas.

### 3.4. Infiltration Rate Monitoring Results

Monthly infiltration rates for PA and PICB pavements were measured at seven locations (F1–F7) as shown earlier in Figure 6. The complete results are listed in Table 3.

**Table 3.** Infiltration rate field monitoring results (data in cm/s).

| Location | F1 | F2 | F3 | F4 | F5 | F6 | F7 |
|---|---|---|---|---|---|---|---|
| | **Porous Asphalt** | | | **Permeable Interlocking Concrete Bricks** | | | |
| **Type of Pavement Date** | South Side | North Side | North Side (Above Drain Ditch) | North Side | North Side (Above Storage Tank) | South Side (Above Storage Tank) | South Side |
| February 2018 | 1.41 | 1.53 | 1.55 | $9.6 \times 10^{-3}$ | $1.6 \times 10^{-3}$ | $6.0 \times 10^{-4}$ | $5.0 \times 10^{-3}$ |
| March 2018 | 2.53 | 1.05 | 1.27 | $5.4 \times 10^{-3}$ | $9.5 \times 10^{-5}$ | $1.5 \times 10^{-3}$ | $6.0 \times 10^{-3}$ |
| April 2018 | 1.26 | 1.11 | 1.35 | $1.1 \times 10^{-2}$ | $3.2 \times 10^{-4}$ | $2.0 \times 10^{-3}$ | $1.1 \times 10^{-3}$ |
| May 2018 | 2.64 | 1.08 | 1.21 | $9.2 \times 10^{-3}$ | $3.7 \times 10^{-4}$ | $4.3 \times 10^{-5}$ | $1.0 \times 10^{-3}$ |
| June 2018 | 1.79 | 0.9 | 1 | $1.8 \times 10^{-3}$ | $3.5 \times 10^{-5}$ | $1.2 \times 10^{-5}$ | $2.0 \times 10^{-4}$ |
| July 2018 | 2.6 | 0.65 | 0.94 | $4.6 \times 10^{-3}$ | $3.9 \times 10^{-4}$ | $4.1 \times 10^{-6}$ | $9.9 \times 10^{-4}$ |
| August 2018 | 2.71 | 0.58 | 0.81 | $2.0 \times 10^{-3}$ | $1.3 \times 10^{-4}$ | $5.4 \times 10^{-5}$ | $4.8 \times 10^{-4}$ |
| September 2018 | 1.89 | 0.45 | 0.77 | $2.7 \times 10^{-3}$ | $1.2 \times 10^{-3}$ | * | $6.4 \times 10^{-4}$ |
| October 2018 | 1.86 | 0.42 | 0.84 | $1.17 \times 10^{-3}$ | $2.66 \times 10^{-4}$ | $1.66 \times 10^{-5}$ | $6.82 \times 10^{-4}$ |
| November 2018 | 1.81 | 0.29 | 0.42 | $1.14 \times 10^{-3}$ | $1.62 \times 10^{-4}$ | $9.59 \times 10^{-5}$ | $2.98 \times 10^{-4}$ |
| December 2018 | 0 | 0 | 0 | $6 \times 10^{-4}$ | $1.91 \times 10^{-4}$ | $2.64 \times 10^{-6}$ | $1.9 \times 10^{-4}$ |
| January 2019 | 0 | 0 | 0 | $2.26 \times 10^{-4}$ | $7.82 \times 10^{-4}$ | * | $3.04 \times 10^{-3}$ |
| February 2019 | 0 | 0 | 0 | $9.99 \times 10^{-3}$ | $1 \times 10^{-3}$ | * | $2.95 \times 10^{-3}$ |
| March 2019 | 0 | 0 | 0 | $1.24 \times 10^{-3}$ | $2.83 \times 10^{-4}$ | * | $1.22 \times 10^{-3}$ |
| April 2019 | 0 | 0 | 0 | $1.44. \times 10^{-3}$ | $1.55 \times 10^{-4}$ | * | $1.12 \times 10^{-3}$ |

\* = No Data.

It should be noted that the porous asphalt bicycle lane was modified in December 2018 to make the surface "smooth" for bike riders. It essentially became impervious at that time.

The results led to the following observations:

Porous asphalt had very high infiltration rates. At location F1, it remained above 1.0 cm/s, or $100 \times 10^{-2}$ cm/s, and for most of the time above 1.5 cm/s or $150 \times 10^{-2}$ cm/s. It should be noted that the government required infiltration rate for porous pavement materials is $1.0 \times 10^{-2}$ cm/s. As a comparison, the Virginia Department of Transportation (VDOT) requires an infiltration rate of $7 \times 10^{-2}$ cm/s for porous asphalt pavements [7].

At locations F2 and F3, the PA infiltration rate started high at 1.5 cm/s and gradually decreased to around 0.5 cm/s, or $50 \times 10^{-2}$ cm/s, after about 10 months. These results indicate a very high infiltration rate for porous asphalt and it remained high even 15 months after installation.

For permeable interlocking concrete bricks (PICB), the infiltration rates were all lower than those for PA. At location F4, the rate started high at $1.1 \times 10^{-2}$ cm/s but declined to less than $0.2 \times 10^{-2}$ cm/s after 11 months, and then increased dramatically, which was probably due to some cleaning work done in the vicinity when the bicycle lane was made impervious. Similar, but not as dramatic, changes of the rate was observed for location F7.

At locations F5 and F6, the rates started low and remained low. One reason could be the uneven compacting work done when the site was constructed.

The infiltration rates for PA and PICB observed at the Taipei site were very similar to those reported by USGS [6] and the Montreal study [8], but quite different from results obtained by USEPA [1]. The inconsistency points to the fact that the actual field infiltration rate of porous pavement depends on many factors, such as installation and site environmental conditions, including underlying soil, runoff sediment load, etc. Additionally, routine maintenance is needed and has to be more frequent for PICB than for PA pavements.

The results discussed above point to the importance of maintenance for porous pavements to perform as expected over time. Since the time variation of infiltration rates differs greatly among different pavement materials, it is proposed that "maintenance schedule" curves be developed based on extensive field or laboratory test data.

In Figure 13, hypothetical infiltration curves for any two assumed porous asphalt materials and one assumed concrete brick material are shown based on available data. The minimum required infiltration is expressed as a horizontal line with, in this case, a value of $1.0 \times 10^{-2}$ cm/s. The intersections of the curves with the required infiltration line give the times maintenance work is needed, i.e., $T_1$, $T_2$, and $T_3$. Such information would be very useful for developing operation/maintenance manuals for porous pavement installations.

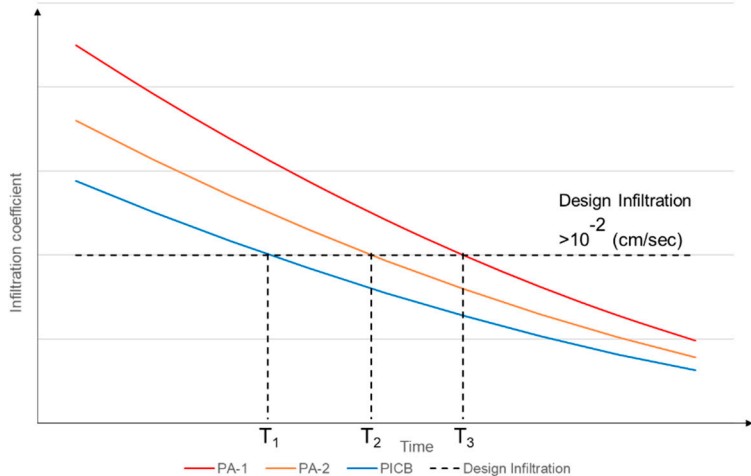

**Figure 13.** Proposed maintenance schedule curves for porous pavements.

### 3.5. Runoff Volume Reduction

The original plan for monitoring runoff reduction by the porous pavement was to measure inflow to, and outflow from, the north/south drainage ditches. However, data collected during many storm events showed larger outflows than inflows. Obviously, there were "unknown" sources of water entering the drain ditches, perhaps due to uneven gradient and/or dumping activities. It was then decided to use model simulation as a means for estimating runoff reduction by porous pavements. The Storm Water Management Model (SWMM), after calibration and verification, as described earlier in the "Materials and Methods" section, was used to calculate runoff reduction rates for the 36 storm events monitored at the site. The results are presented in Table 4 below.

**Table 4.** Runoff reduction rate.

| Date | Total Rainfall (mm) | Rainfall Duration | (North/South) Reduction Rate (%) |
|---|---|---|---|
| 12 April 2018 | 35.5 | 01:10 | 21/35 |
| 16,17 April 2018 | 17.8 | 08:40 | 47/45 |
| 8 May 2018 | 26.6 | 04:00 | 41/54 |
| 30 May 2018 | 18.2 | 00:30 | 54/52 |
| 10,11 July 2018 | 126.8 | 16:10 | 22/32 |
| 21 July 2018 | 26.6 | 07:20 | 44/42 |
| 24 July 2018 | 27.8 | 02:00 | 43/41 |
| 11 August 2018 | 30.2 | 07:50 | 45/46 |
| 12 August 2018 | 45.8 | 06:50 | 44/46 |
| 17 August 2018 | 31.2 | 01:40 | 45/55 |
| 20 August 2018 | 13.0 | 00:40 | 45/47 |
| 30 August 2018 | 54.4 | 09:20 | 45/45 |
| 1 September 2018 | 22.6 | 01:10 | 44/42 |
| 7 September 2018 | 39.6 | 03:50 | 29/26 |
| 8 September 2018 | 152.0 | 06:00 | 17/16 |
| 9 September 2018 | 35.5 | 17:20 | 41/41 |
| 15,16 September 2018 | 17.8 | 27:30 | 44/43 |

**Table 4.** *Cont.*

| Date | Total Rainfall (mm) | Rainfall Duration | (North/South) Reduction Rate (%) |
|---|---|---|---|
| 25–27 September 2018 | 26.6 | 35:00 | 44/44 |
| 10,11 October 2018 | 16.8 | 13:05 | 45/45 |
| 11,12 October 2018 | 25.8 | 17:45 | 38/36 |
| 16,17 October 2018 | 30.8 | 11:00 | 50/47 |
| 1/2 November 2018 | 21.8 | 42:10 | 55/55 |
| 23,24 December 2018 | 41.2 | 30:40 | 44/43 |
| 16,17 January 2019 | 21.2 | 25:00 | 77/77 |
| 23,24 February 2019 | 57.4 | 22:40 | 43/42 |
| 6–10 March 2019 | 89.4 | 84:20 | 41/40 |
| 25 March 2019 | 15.0 | 06:10 | 43/42 |
| 29,30 March 2019 | 24.0 | 05:20 | 39/28 |
| 11 April 2019 | 15.6 | 08:30 | 47/45 |
| 15,16 April 2019 | 29.0 | 24:00 | 43/42 |
| 20,21 April 2019 | 40.6 | 10:40 | 42/41 |
| 22 April 2019 | 23.8 | 05:00 | 44/42 |
| 1 May 2019 | 46.6 | 17:10 | 43/42 |
| 2,3 May 2019 | 14.4 | 20:30 | 44/43 |
| 6,7 May 2019 | 24.2 | 27:10 | 43/43 |
| 9,10 May 2019 | 14.0 | 08:20 | 43/43 |

The runoff reduction rate varied from 16% to 77%, with the south side showing slightly higher rates (averaged 40.6% to 34.6% for the north side), which could be attributed to higher infiltration rates at F7. The overall average rate of runoff reduction was therefore 37.6%, which is similar to that reported by [8] for permeable interlocking concrete pavement.

Data generated by the present study and those found in the literature all indicate that the runoff reduction rate is closely related to rainfall characteristics such as intensity, duration, and total volume of rain. Regression analyses were made relating the reduction rate to various rainfall parameters such as total rainfall volume per specific period of time, intensity, and time-variation of rainfall. It was found that the reduction rate vs. total amount of rainfall per hour had the highest correlation, as depicted in Figure 14.

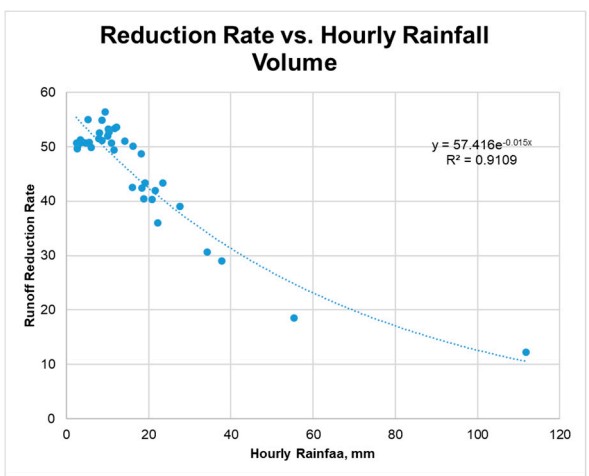

**Figure 14.** Runoff reduction rate vs. total hourly rainfall.

## 4. Conclusions

The paper presents results of a 15-month field test of two types of porous pavement performance in Taipei City. A porous asphalt (PA) bicycle lane and a permeable interlocking concrete brick (PICB) pedestrian walkway were monitored for surface temperatures at 10 min intervals during storm events and after long dry periods. Infiltration rates were measured monthly. In addition, two underground

storage tanks at the site were monitored, and the performance on runoff reduction by the porous pavements was simulated by using a storm water management model.

Porous pavements help reduce surface temperature, especially during high air temperature periods. In summer, during storm events, PA showed a maximum temperature drop of 3.9 °C and for PICB, 6.6 °C. In winter when the air temperature was low, the temperature drop was 0.7 °C and 0.6 °C for PA and PICB, respectively. The temperature drop was much more pronounced during long dry periods. A drop of 14.9 °C was observed for PA and 9.4 °C for PICB.

For PA, the infiltration rate was high and remained high (above $1.0 \times 10^{-2}$ cm/s) at location F1 during the 15 month monitoring period. At two other locations the rates started high but gradually decreased to about $0.2 \times 10^{-2}$ cm/s after 10 months. For PICB the rates were all low. Clogging was a problem, but infiltration rates would improve with maintenance.

The storage tanks were full only twice when the rainfall amounts were large and extra-large (>100 mm/d). The runoff reduction averaged 35% (north side) to 41% (south side) according to SWMM simulation results. As expected, the runoff reduction rate decreased rapidly when rainfall amount and intensity increased, e.g., the rate dropped to about 10% when rainfall intensity exceeded 100 mm/h (Figure 14). This suggests that LID or green infrastructure practices, most of which are designed for small storms (e.g., 1 to 2-year return periods), could only control smaller storm events, which was the original intent during their development. For larger storms, a combination of green and grey infrastructure is needed.

In summary, the results of the 15 month field tests at the Taipei City site clearly demonstrated the environmental benefits of porous pavements in terms of (a) surface runoff peak and volume reduction; (b) rainfall water harvesting; (c) groundwater replenishment; (d) urban pavement surface temperature reduction that could help mitigate the urban heat-island effect; and (e) evaporation of infiltration runoff into the pavement material, which could help reduce the "black ice" potential in colder areas. On the other hand, assessment of the economic benefits is a complex process and is beyond the contractual scope of the present study. The study period should necessarily be long enough to allow a full life-cycle cost analysis (LCCA) of the porous pavement installation. Compared to traditional pavement, the porous pavements required an additional 10%–15% construction cost at the Taipei test site. However, a long-term monitoring effort is needed in order to conduct a life-cycle cost analysis.

**Author Contributions:** Y.-Y.C., data curation, formal analysis; S.-L.L., formal analysis, supervision; C.-C.H., data curation, methodology, formal analysis; J.-Y.L., funding acquisition, supervision, formal analysis, validation; S.L.Y., conceptualization, formal analysis, writing – original draft, review and editing.

**Funding:** This research was funded by the Taipei City Government, Taiwan, grant number – H-106-04-106230.

**Conflicts of Interest:** The authors declare no conflict of interest. The funders had no role in the design of the study; in the collection, analyses, or interpretation of data; in the writing of the manuscript, or in the decision to publish the results.

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
