# Peer review of "Field Testing of Porous Pavement Performance on Runoff and Temperature Control in Taipei City"

_water, doi:10.3390/w11122635_

Round 1
Reviewer 1 Report
This paper reports a small experiment on the effect of paved surfaces in stormwater runoff. It examines infiltration through intelicking brick and porous asphalt pavements in comparison with noirmal road surfaces. The secen setting and experimentasl swt-up information are good, but some details are mssing. For example, it was not clear how the infitration rate was measured and how the pavement surface temperature were measured. In some cases there are mentions of sources from whis=ch measurement techniques were derived, but at least the principles of the methods used shoud be explained so that the reader can judge whether they are appropriate for the experiment.
Thee is no data to indicate how typical the study period was of the long-term weather patterns in the area. I have concerns that the authors may use models and techniques from other parts of the world that may be based on assumptions that reflect different climatic conditions and different types of terrain.
I have indicate on th attached annotated pdf of the paer where I have concerns.

Reviewer 2 Report
The manuscript presents the results of field testing of porous pavement performance on temperature, infiltration rate and runoff. The topic is of great current interest. The overall structure of the manuscript is correct but individual chapters are holding serious flaws and must be revised.
The Introduction chapter needs some adjustment. Starting with the style of references. Authors must conform to the journal's norm to cite using numbers in brackets e.g. [1]and list them in the order of their appearance in the text. The statements in L38-44 require providing references. Figure 1 is not relevant and should be removed.
The Materials and Methods chapter is missing some important information. Section 2.1 should be renamed to ‘Site description’ and supplemented with the map of site location. The Authors should consider to replace Figures 2 and 3 with the vertical cross-sections revealing the dimensions of storage tanks, layers and materials used. In section 2.2 more details about setup of measurements are requested e.g. how the flow is measured, what instruments were used to measure infiltration rates, what was the vertical position of the temperature sensors. The flow paths of runoff water should be presented and explain what areas contributes to each drain ditch and storage tank. Also all porous materials should be characterize with specific gravity, porosity etc.
The Results and Discussion chapter should be rewritten in some parts. Two statements about maximum rainfall in L144-145 are not consistent if the same units are used. In section 3.2 to enhance discussion please provide the highest levels of water table and max amplitude of water table change in the whole monitoring period. Regarding Figure 5 please explain what dates were chosen and explain why these particular dates. What was the time step of presented results? Is there any correlation between water table elevation and fullness of storage tank? Check the number of months of the monitoring period (L158, L248). In Table 2 please replace N and S with North and South respectively and rename the ‘Rate of storage’ with ‘Fullness’. In section 3.3 please explain why the results of temperature measurements are presented? Are they correlated with run-off or infiltration rate? If there is no good justification then this section is out of journal’s scope. There is inconsistency in time span for measurements of infiltration (Table 3) and precipitation (Table 1). Figure 10 duplicates data presented in Table 3. In L275 is written that there were ‘two porous asphalt materials’. Please describe the differences in ‘Materials and Methods’. If in section 3.5 the model was used to calculate the runoff reduction rate then the model should be described in ‘Materials and Methods’. Please explain also what was the reference value to calculate the reduction. Please avoid repetition of Table 1 in Table 4.
The chapter Conclusions should not be only a summary of analysis but should contain the main findings and underline main economic and environmental benefits of using these materials if they are applied at larger scale.
References list contains only 2 scientific papers. The Authors are advised to do additional literature search.
Round 2
Reviewer 1 Report
The improvements demonstrated in the revised paper are good. I just wish to see the limited flood attenuation benefits of porous pavements in extreme storms emphasised (Figure 13 comments).

Reviewer 2 Report
I am thankful to the Editor for giving me the opportunity to review the revised version of the manuscript. Comparing to the original submission it has improved significantly. However the Authors did not addressed all the comments appropriately. Therefore please consider once again the following specific comments:
Not all citations were revised according to the journal’s norm. See L33 There are two drain ditches in the test site and the flow is measured only in one. Please clarify in the text which surface contribute to the ditch with flow measurements. Side walk, bicycle lane or both? I do not understand the sentence in L157-158. Does it mean the storage tanks collect runoff or some other water of different origin? Please provide units for Elevation on Fig. 7 and bottom elevation of the storage tanks. I asked for justification of presenting the temperature results because they seemed to me not related to water. The Authors answered ‘the major thrust of this research effort is to test whether porous pavements could help alleviate the urban heat-island effect’. I still think that this part is not relevant for this journal. Why not describe them in separate paper and submit to other journal? Please state clearly that Fig. 12 presents only examples of infiltration curves not real ones. In reply to my comments the Authors say: ‘We added a paragraph to describe to stormwater management model SWMM used in the study.’ Unfortunately I could not find this paragraph in revised manuscript. In my opinion the SWMM model set-up should be described in details with the results of calibration and validation.Author Response
Please see the attachment
